# Income-based differences in healthcare utilization in relation to mortality in the Swedish population between 2004–2017: A nationwide register study

**Pär Flodin**⊙*, **Peter Allebeck**⊙, **Ester Gubi**⊙, **Bo Burström**⊙, **Emilie E. Agardh**

Department of Global Public Health, Karolinska Institutet, Stockholm, Sweden

* parflodin@gmail.com

## Abstract

**Data Availability Statement:** Data cannot be shared publicly because of personal integrity. Data are available from the Swedish National Board of Health and Welfare and Statistics Sweden, for researchers who meet the criteria for access to confidential data. Data on primary care utilization

### Background

Despite universal healthcare, socioeconomic differences in healthcare utilization (HCU) persist in modern welfare states. However, little is known of how HCU inequalities has developed over time. The aim of this study is to assess time trends of differences in utilization of primary and specialized care for the lowest (Q1) and highest (Q5) income quantiles and compare these to mortality.

### Methods and findings

Using a repeated cross-sectional register-based study design, data on utilization of (i) primary; (ii) specialized outpatient; and (iii) inpatient care, as well as (iv) cause of death, were linked to family income and sociodemographic control variables (for instance, country of origin and marital status). The study sample comprised all individuals 16 years or older residing in Sweden any year during the study period and ranged from 7.1 million in year 2004 to 8.0 million year 2017. HCU and mortality for all disease as well as for the 5 disease groups causing most deaths were compared for the Q1 and Q5 using logistic regression, adjusting for sex, age, marital status, and birth country. The primary outcome measures were adjusted odds ratios (ORs), and regression coefficients of annual changes in these ORs log-transformed. Additionally, we conducted negative binominal regression to calculate adjusted rate ratios (RRs) comparing Q1 and Q5 with regard to number of disease specific healthcare encounters ≤5 years prior to death. In 2017, for all diseases combined, Q1 utilized marginally more primary and specialized outpatient care than Q5 (OR 1.07, 95% CI [1.07, 1.08]; $p$ < 0.001, and OR 1.04, 95% CI [1.04, 1.05]; $p$ < 0.001, respectively), and considerably more inpatient care (OR 1.44, 95% CI [1.43, 1.45]; $p$ < 0.001). The largest relative inequality was observed for mortality (OR 1.78, 95% CI [1.74, 1.82]; $p$ < 0.001). This pattern was broadly reproduced for each of the 5 disease groups. Time trends in HCU inequality varied by level of care. Each year, Q1 (versus Q5) used more inpatient care and suffered increasing mortality rates. However, utilization of primary and specialized outpatient care increased more

can be requested from the following 17 regional database owners: https://www.regionstockholm.se/om-regionstockholm/forskning-och-innovation/centrum-for-halsodata/fragor-och-svar-for-forskare-om-halsodatabaser/ https://regionuppsala.se/samverkanswebben/it-service-och-fastighet/it-system/sas-viya/bestallning-av-data-for-forskning/ https://www.regionostergotland.se/ro/det-har-gor-vi/forskning/for-dig-som-forskar/forskarservice-och-infrastruktur/ansokan-om-registeruttag-fran-vardregister fouenheten@rjl.se primarvardsforvaltningen@ltkalmar.se statistikehalsamit@gotland.se, region@regionblekinge.se https://vardgivare.skane.se/kompetens-utveckling/forskning-inom-region-skane/utlamnande-av-patientdata-samradkvb/ https://www.vgregion.se/statistik-analysportalen/bestalla-statistik-och-data/ https://www.regionvarmland.se/ https://www.regionorebrolan.se/sv/forskning/kontakt-och-organisation/Forskning–och-utbildningsledning/ forskning@regionvastmanland.se https://www.regiondalarna.se/plus/forskning/personuppgifter-till-forskning/ https://www.regiongavleborg.se/samverkanswebben/halsa-vard-tandvard/samverkan-och-avtal/halsovalet/Kontakt/ https://www.rvn.se/sv/Om-regionen/regionens-organisation/patientsakerhet-utveckling-och-forskning/Forskning-och-utbildning/fouu-och-folkhalsa/Personal/ Beslutsstod@regionjh.se forskning@norrbotten.se.

**Funding:** This research received funding from Swedish Research Council for Health, Working Life and Welfare (website:https://forte.se/en/), Grant number: DNR: 2021-00176 (to EA). The funders had no role in study design, data collection and analysis, decision to publish, or preparation of the manuscript.

**Competing interests:** The authors have declared that no competing interests exist.

**Abbreviations:** HCU, healthcare utilization; NBHW, National Board of Health and Welfare; OR, odds ratio; RR, rate ratio; SES, socioeconomic status.

among Q5 than in Q1. Finally, group differences in number of healthcare encounters ≤5 years prior to death demonstrated a similar pattern. For each disease group, primary and outpatient care encounters were fewer in Q1 than in Q5, while inpatient encounters were similar or higher in Q1. A main limitation of this study is the absence of data on self-reported need for care, which impedes quantifications of HCU inequalities each year.

## Conclusions

Income-related differences in the utilization of primary and specialized outpatient care were considerably smaller than for mortality, and this discrepancy widened with time. Facilitating motivated use of primary and outpatient care among low-income groups could help mitigate the growing health inequalities.

## Author summary

### Why was this study done?

- In recent decades, Sweden has witnessed a rise in income inequalities, accompanied by shifts in the sociodemographic composition of the population and transformations of the healthcare system.

- Previous studies show that high-income groups utilize specialized outpatient care more than low-income groups (after adjusting for need), yet little is known about the development of inequalities in the utilization of primary, outpatient, and inpatient care over time.

### What did the researchers do and find?

- We linked income and sociodemographic data to data on utilization of primary, outpatient, and inpatient care, as well as to mortality, to track differences in care utilization between the lowest and the highest income quintile in the Swedish adult population for more than a decade.

- We found that the lowest income quintile utilized a decreasing proportion of primary and outpatient care, despite having increasing mortality rates.

- The disparities between inequalities in healthcare utilization (HCU) and mortality were most pronounced for neoplasms and chronic respiratory diseases, while being less prominent for neurological disorders.

### What do these findings mean?

- By comparing the trends in income-related differences in HCU with trends in mortality inequalities, we here provide evidence of increasing inequalities in utilization of primary and outpatient care over time.

- To deliver healthcare in proportion to needs and to ensure efficient use of healthcare resources, the health sector should promote motivated utilization of primary and specialized care among low-income groups.

- A main limitation of this study is the absence of self-reported healthcare needs, restricting our ability to quantify inequalities in HCU.

- Future studies should assess to what extent group differences in care utilization explains inequalities in mortality.

## Introduction

Low socioeconomic status (SES) is associated with higher mortality rates and more years lived with disability and disease [1]. Yet, in most low- and middle-income countries, low SES groups utilize less healthcare and of worse quality [2], a phenomenon referred to as the "inverse care law" [3]. Despite universal healthcare, socioeconomic differences in healthcare utilization (HCU) persist also in many modern welfare states, where low SES groups commonly use more healthcare but not in proportion to their additional needs (sometimes called the "disproportionate care law") [2,4]. Similar to many other European countries, the Swedish law mandates that the healthcare should be provided in proportion to the need of individuals [5]. However, the degree to which this has been accomplished is not conclusively determined.

In a recent systematic review of 57 studies from high-income countries, Lueckmann and colleagues [4] report that in the majority of studies, higher SES was associated with higher utilization of specialized outpatient care for a given self-reported health status. The findings of primary care utilization were more heterogeneous, and the authors concluded that inequities were more common in specialized outpatient care compared to primary care. Hence, Lueckmann and colleagues [4] pointed out that research on socioeconomic inequalities in HCU should differentiate between primary care and specialized outpatient care, since the results differ greatly according to the type of service.

To evaluate the effects of public policy and healthcare reforms on inequality in HCU and mortality, analysis of time trends are informative [6]. Although Sweden was successful in reducing inequalities, for instance, by providing universal healthcare and free education, relative inequalities in mortality have increased since 1990 [7]. The Swedish healthcare system has, during the same period, undergone major changes, such as the marketization of primary care (spurred by the Primary Health Care Choice Reform in 2010) and a recent increase in virtualization of primary care. Although previous research indicates that marketisation increased inequalities in HCU [8], the evidence is sparse and incomplete [9]. Similarly, studies of trends in inequalities in utilization of specialized care show mixed results [10,11], and updated, large-scale studies are lacking. Furthermore, the extent to which trends in socioeconomic differences in HCU at various levels of care compare to trends in mortality inequalities has not been thoroughly studied.

The Scandinavian health registers provide unique opportunities to investigate this. In Sweden, national population healthcare registers are regularly updated [12], and microdata on HCU and death can be linked to social databases through civic registration numbers. This enables analysis of socioeconomic differences and utilization of primary and specialized outpatient and inpatient care as well as mortality for the general population over time. Unfortunately, Sweden is still lacking a national register of primary care utilization. To our knowledge,

this is the first study studying income-related differences at all levels of care at the national level.

The aim of this study is to assess time trends of income-related differences in use of different levels of healthcare in relation to mortality between the years 2004 and 2017. To do this, we compared the lowest (Q1) with the highest income quintile (Q5) with regard to the utilization of primary care, specialized outpatient care, and inpatient care. To facilitate interpretation of income-related group differences in HCU, we included mortality as a comparative outcome measure. We studied each of the 5 disease groups that account for the largest number of deaths in Sweden 2017 (cardiovascular diseases, neoplasms, neurological disorders, chronic respiratory diseases, and diabetes and chronic kidney diseases), as well as all-disease. Specifically, we addressed the following questions: (1) How do low- and high-income groups differ with regard to HCU and mortality in 2017 and over time? (2) To what extent do income-related differences in HCU and mortality vary across disease groups? (3) How do the low- and high-income groups differ in numbers of healthcare encounters ≤5 years prior to death?

## Methods

The study is reported according to the Strengthening the Reporting of Observational Studies in Epidemiology (STROBE) guideline (S1 Checklist). No study protocol was submitted prior to submission.

### Study population and data sources

The study population comprised all individuals 16 years and older living in Sweden, any year between 2004 and 2017, that were registered in the Longitudinal Integrated Database for Health Insurance and Labour market Studies [13] by Statistics Sweden (Table 1). Data on specialized out- and inpatient HCU (from the National Patient Register) and mortality (from the Cause of Death Register) were obtained from the Swedish National Board of Health and Welfare (NBHW). Since primary care data are not available at the national level, we contacted the 21 Swedish counties maintaining electronic records of diagnoses in primary care. Seventeen counties provided time series covering one or more years (S1 Table). Whereas the national health registers provided by NBHW have well-documented and high population coverage [12], primary care data lacked estimates of coverage rates, which likely varied by region and over time. Hence, crude rates of primary care utilization could partly be affected by increasing coverage rates and should be interpreted with caution. For disease-specific analyses, we selected the five disease categories causing the highest number of deaths in Sweden in 2017, which together were responsible for 84% of all deaths [14]. The disease groups were (with percentage of the total number of deaths in parentheses): cardiovascular diseases (38%), neoplasms (28%), neurological disorders (8%), chronic respiratory diseases (5%) and diabetes and chronic kidney diseases (pooled) (4%). Diseases were classified using the tenth version of the WHO International Classification of Diseases (ICD-10) codes in accordance with the disease groups defined by the Global Burden of Disease project (S2 Table). Whereas NBHW define specialized care and underlying cause of death by ICD-10, diagnoses in primary care were classified using ICD-10-P which we mapped to ICD-10. Since ICD-10-P is less detailed than the ICD-10, certain disease groups in the primary care contain marginally broader disease categories than the GBD target groups.

### Income measures

The study population was grouped by income ranks based on equivalised disposable family income. Family income referred to wages, capital returns, self-employment, pensions, and

**Table 1. Characteristics of the study populations.**

| Year(s) | N | | | Age (SD) | | | Proportion born in Sweden (%) | | | Proportion married (%) | | | Income ratio[a] (Q5/Q1) |
|---|---|---|---|---|---|---|---|---|---|---|---|---|---|
| | Tot. pop. | Q1 | Q5 | Tot. pop. | Q1 | Q5 | Tot. pop. | Q1 | Q5 | Tot. pop. | Q1 | Q5 | |
| **Population denominators for analyses of outpatient care, inpatient care, and mortality** | | | | | | | | | | | | | |
| All years | 105,343,853 | 21,075,466 | 21,063,689 | 48.5 (19.4) | 48.5 (19.4) | 48.5 (19.4) | 84.5 | 65.2 | 92.2 | 41.8 | 34.1 | 47.3 | 3.5 |
| 2017 | 8,001,909 | 1,600,742 | 1,600,073 | 48.7 (19.6) | 48.7 (19.6) | 48.7 (19.6) | 81.2 | 56.6 | 91.1 | 41.0 | 32.8 | 46.7 | 4.3 |
| 2011 | 7,625,859 | 1,525,750 | 1,524,841 | 48.1 (19.6) | 48.1 (19.6) | 48.1 (19.6) | 84.6 | 65.2 | 92.4 | 41.4 | 33.3 | 46.7 | 3.5 |
| 2004 | 7,086,141 | 1,417,981 | 1,416,834 | 48.5 (19.1) | 48.5 (19.1) | 48.5 (19.1) | 86.9 | 72.4 | 92.7 | 43.1 | 37.0 | 48.4 | 3.1 |
| **Population denominators for analyses of primary care** | | | | | | | | | | | | | |
| All years | 77,812,350 | 15,598,476 | 16,678,951 | 48.2 (19.4) | 47.9 (19.21) | 48.2 (19.3) | 82.9 | 61.5 | 91.6 | 41.5 | 34.1 | 47.0 | 4.2 |
| 2017 | 7,142,796 | 1,436,285 | 1,456,728 | 48.6 (19.6) | 48.69 (19.6) | 48.8 (19.6) | 80.7 | 56.1 | 90.8 | 40.8 | 32.7 | 46.5 | 4.4 |
| 2011 | 5,841,894 | 1,178,215 | 1,239,690 | 47.8 (19.5) | 47.6 (19.4) | 47.9 (19.5) | 83.6 | 62.9 | 92.0 | 41.0 | 33.5 | 46.4 | 4.2 |
| 2004 | 4,101,156 | 803,496 | 962,576 | 47.8 (18.9) | 46.9 (18.5) | 48.0 (18.9) | 83.95 | 65.0 | 91.9 | 42.8 | 37.6 | 47.9 | 4.3 |

The study populations analyzed for outpatient care, inpatient care, and mortality (top), and primary care (bottom). For the analyses of outpatient care, inpatient care, and mortality, the study population encompasses the entire Swedish population aged 16 years and older, excluding a marginal proportion of individuals (<2%) with incomplete demographic data or nonpositive family incomes. For the primary care analyses, only the subset of citizens from counties with available primary care data was considered.

N, number of included individuals; SD, standard deviation; Tot. pop., total population, i.e., all 5 income quantiles summed; Q1, lowest income quantile; Q5, highest income quantile; All years, 2004–2017, summed (for N) or averaged using weighted means (for age, the proportion born in Sweden, proportion married, and income ratio).

[a]Income ratios were calculated by dividing Q5 family income by that of Q1.

social benefit, after taxes were deducted (see S1 Appendix for further information on the income data). For each studied year, an individual's family income was averaged across the 1 to 5 preceding years, and the resulting variable was used for income ranking. Individuals with a family income of zero or below (<2% of the population) were excluded from analysis due to the heterogeneity of this group [15]. Income groups were defined as quintiles of time averaged family income, stratified by sex and birth year. Thus, at a given year, each income quintile contained approximately the same number of individuals and were balanced with regard to sex and age compositions.

## Statistical analysis

Crude rates of HCU per 100,000 person/year were calculated for each disease group and for each of the 3 levels of care. Only main diagnoses were considered. Mortality rates were based on underlying cause of death. For main analyses, we used the number of unique individuals, where an individual was counted once per disease group, calendar year, and type of data source (i.e., primary care, outpatient, inpatient, or death). Given the varying coverage rate of primary care register data, the counties contributing to the population denominator varied

across the years, why it ranged from 4.1 million individuals in 2004 to 7.1 million in 2017. The population denominator used for the national health register data ranged from 7.1 million individuals in 2004 to 8.0 million in 2017 (Table 1).

We conducted multiple logistic regressions to calculate adjusted odds ratios (ORs) of HCU or death, comparing the lowest (Q1) and highest (Q5) income quintiles. The model adjusted for country of origin, civil status, sex, age, and age squared (see S1 Appendix for details). To ensure consistency, we also did a sensitivity analysis using the primary care study population across all analyses.

To infer time trends in ORs, we regressed log-transformed OR on time, using year as a continuous independent variable. To account for uncertainty of the OR, we conducted bootstrapping analysis by simulating new time series (1,000 resamples with replacements) drawn from a normal distribution with mean and standard deviation equal to the log-transformed ORs. For each simulated time series, we performed linear regressions (GLM) and inferred 95% CI from the sampling distribution of the estimated beta coefficients. The linear model was deemed the most parsimonious and suitable choice, given the limited number of time points and after visually examining the trends.

We then tested if the income groups differed in numbers of healthcare visits during a 5-year period prior to death. More specifically, for each year and each disease group, we ran a negative binominal regression, regressing the number of cause of death specific healthcare encounters on income quintiles. The resulting rate ratios (RRs) were adjusted for age, age squared, sex, marital status, and country of origin.

Finally, we performed 2 sets of sensitivity analyses. First, we replicated all preceding analyses, using unadjusted models (subsequently referred to as Model 2), excluding any control variables. Second, we reexecuted all original analyses but, in addition to the original control variables, incorporated 21 categorical county variables to also adjust for county-specific effects (referred to as Model 3). Further details on the original Model 1, the unadjusted Model 2, and the county adjusted Model 3 can be found in the S1 Appendix.

Analyses were carried out in SAS 9.4 and in Python 3.6 using standard data science packages such as *statsmodels*. All code is available at https://github.com/parflo/TrendsInMortalityHCUInequality.git.

The study was approved by the Stockholm Regional Ethical Review Board (DNR: 2018/1339-31/5, 2018/2292-32) and the Swedish Ethical Review Authority (2019–02185, 2021–00657 and 2022-03111-02).

## Results

For the analyses of mortality and utilization of inpatient and outpatient care, we investigated 105.3 million person-years at risk. For primary care, we had access to data on 77.8 million person-years at risk. Outcomes in the cohort amounted to 1.3 million deaths, 9.9 million year-unique individuals in inpatient care, 38.4 million in outpatient care, and 45.9 million in primary care (Tables 1 and S3).

Fig 1 shows OR for utilization of primary care, outpatient care, inpatient care, and mortality for each of the disease categories in 2017. Considering all-disease, those with lowest income (Q1) utilized slightly more primary care (OR = 1.07, 95% CI [1.07, 1.08]; $p < 0.001$) and outpatient care (OR = 1.04, 95% CI [1.04, 1.05]; $p < 0.001$) compared to those with the highest income (Q5). However, Q1 utilized considerably more inpatient care (OR = 1.44, 95% CI [1.43, 1.45]; $p < 0.001$) and had even higher death rates (OR = 1.78, 95% CI [1.74, 1.82]; $p < 0.001$) (Fig 1 and S4 Table).

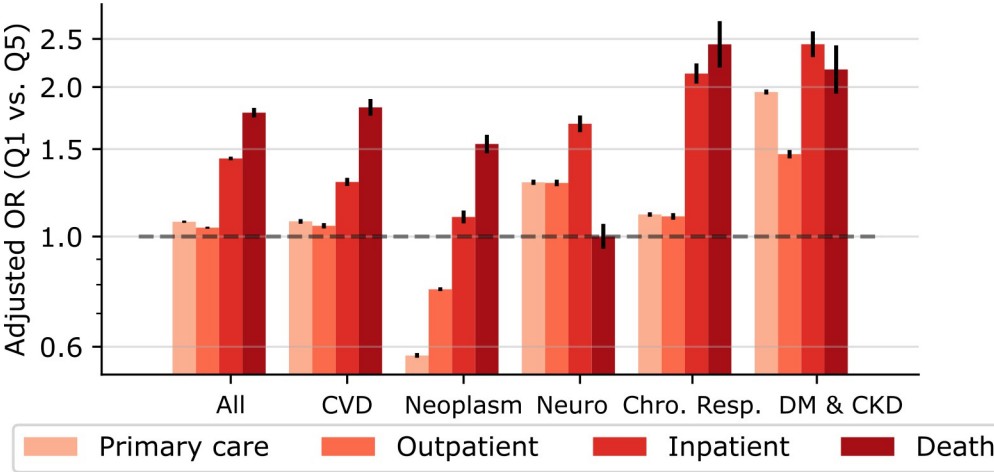

**Fig 1. Adjusted ORs of HCU and mortality comparing the lowest with the highest income quintile, in year 2017.**
ORs are adjusted for sex, age, age squared, country of birth, and civil status. Unadjusted results are available in S4 Table and yielded largely similar results. All, any ICD-10 code; CVD, cardiovascular diseases; Neuro, neurological disorders; Chro. Resp., chronic respiratory diseases; DM & CKD, diabetes and kidney diseases; HCU, healthcare utilization; OR, odds ratio.

Similar patterns of income-related HCU and mortality were observed for each investigated disease. For both cardiovascular diseases and neoplasms, ORs were lowest for utilization of primary and outpatient care and were considerably larger for inpatient care. The largest relative difference was found for mortality. Interestingly, use of primary and outpatient care due to neoplasms were even lower in Q1 than in Q5 (primary care: OR = 0.58, 95% CI [0.57, 0.58]; $p < 0.001$; outpatient care: OR = 0.78, 95% CI [0.78–0.79]; $p < 0.001$) (Fig 1 and S4 Table).

Fig 2 shows time series of rates of HCU and mortality for Q1, Q5, and for the total study population. For all the 6 disease categories, the yearly rates of unique individuals in primary and outpatient care are increasing. Rates of inpatient care plateaued around 2012 for most disease categories and decreased by the end of the study period. All-disease mortality rates for Q5 and the total population declined throughout the study period but stayed relatively unchanged for Q1. Cardiovascular diseases contributed most to decreasing mortality rates in the total population (S3 Table). Time series of rates of HCU and mortality for each of the 5 income quantiles are shown in S1 Fig.

Fig 2 also shows time series of adjusted OR comparing Q1 with Q5. We observe an overall pattern of slightly decreasing ORs in primary and outpatient care, but increasing ORs in inpatient care and mortality, for most disease categories. Nearly identical results were obtained from the sensitivity analysis in which we used the same population for specialized care and mortality as was used for primary care (S2 Fig).

Zooming in on the time trends of OR, Fig 3 displays the beta estimates (i.e., the slope) of log-transformed ORs regressed on time, which denotes the annual rate of change in the relative differences between Q1 and Q5. Time trends of ORs for all-disease followed a similar pattern as the one observed for 2017. Thus, for each year, Q1 (relative to Q5) displayed increasing rates of inpatient care, and even larger increase in mortality rates, but decreasing rates of primary and, especially, outpatient care. For most of the specific disease categories, including cardiovascular diseases and neoplasms, we observed a similar pattern of HCU and mortality trends (Fig 3 and S4 Table).

Fig 4 shows income-related difference in number of disease-specific healthcare encounters within 5 years prior to death. In 2017, Q5 had significantly higher rates of both primary and

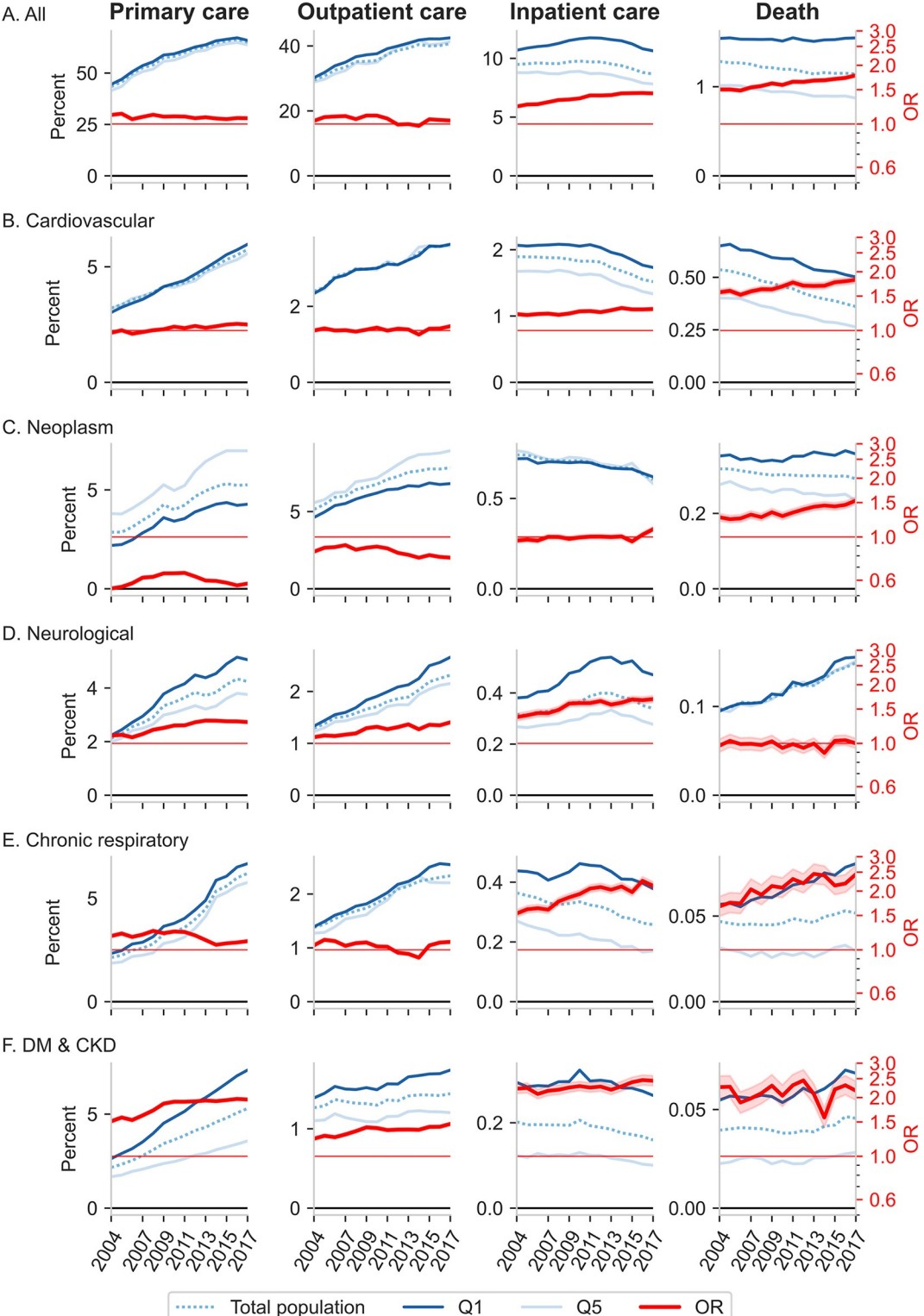

**Fig 2. Time courses (2004–2017) of rates (cases per 100) and adjusted ORs of HCU and mortality.** HCU (yearly rates of unique individuals) and mortality rates, by income group and for the total study population, are shown in blue colors. Rates are presented in percentages for display purposes, and corresponding numerical rates per 100,000 are found in S3 Table. Adjusted ORs (95% CI) comparing Q1 with Q5 are in red, and shaded areas denote 95% CI. ORs are adjusted for sex, age, age squared, country of birth, and civil status. Unadjusted results are available in S4 Table. Red horizontal grid lines denote

OR = 1, and the black grid lines denote the zero levels of rates. All, any ICD-10 code; CVD, cardiovascular diseases; Neuro, neurological disorders; Chro. Resp., chronic respiratory diseases; DM & CKD, diabetes and kidney diseases; HCU, healthcare utilization; OR, odds ratio.

outpatient care utilization across all 5 disease categories. Notably, for neoplasms, Q1 had only half the number of primary care encounters compared to Q5 (RR 0.50, 95% CI [0.40, 0.61]; $p < 0.001$) and a slightly less reduced number in outpatient care (RR 0.83, 95% CI [0.79, 0.88]; $p < 0.001$). Group differences in outpatient care were largest for neurological disorders (RR 0.50, 95% CI [0.45, 0.56]; $p < 0.001$). For inpatient care, RRs did not significantly differ from 1 in any disease, except from all-disease, where Q1 displayed higher rates (RR 1.17, 95% CI [1.11, 1.23]; $p < 0.001$).

The post hoc sensitivity analysis of unadjusted ORs and unadjusted RRs did not alter the original results in any way that significantly impacted the conclusions, as detailed in S4 and S5 Tables. However, in the sensitivity analyses where county was controlled for (i.e., Model 3), the OR for primary care was generally higher than in the original analysis. The strongest modifications were the marked reductions in income-related differences in number of primary healthcare encounters ≤5 years prior to death: In 2017, all group differences were erased in each of the 6 disease categories, apart from neurological disorders (S5 Table).

## Discussion

In this study, we investigated time trends of income-stratified HCU, along with mortality as a comparative outcome. Whereas income-related differences were small for primary and outpatient care, they were larger for inpatient care. This was true for all the investigated disease groups. However, except for neurological disorders and diabetes and chronic kidney diseases, the largest inequalities were observed for mortality. Considering all-disease at the end of the study period (2017), the lowest income quintile had 7% higher odds of utilizing primary care, 4% higher odds of utilizing specialized outpatient care, 44% increased odds of hospitalization, and 78% higher odds of dying. The largest discrepancy between group differences in utilization

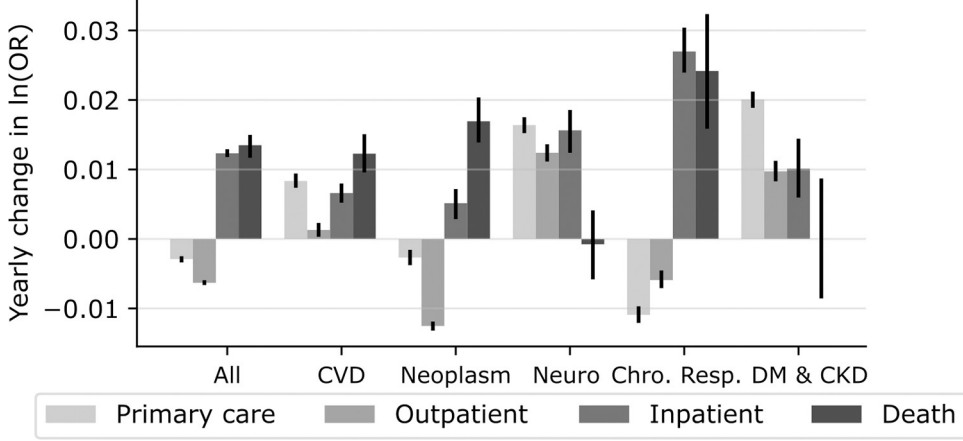

**Fig 3. Yearly change in log-transformed adjusted ORs.** Bars show the beta-coefficients (i.e., rates of change or "slopes") estimated by linear regression of the log-transformed time-series of OR regressed on years (2004–2017). Error bars denotes 95% CI. Analyses are adjusted for sex, age, age squared, country of birth, and civil status. Numerical values for both adjusted and unadjusted are found in S4 Table. All, any ICD-10 code; CVD, cardiovascular diseases; Neuro, neurological disorders; Chro. Resp., Chronic respiratory diseases; DM & CKD, Diabetes and kidney diseases; OR, odds ratio.

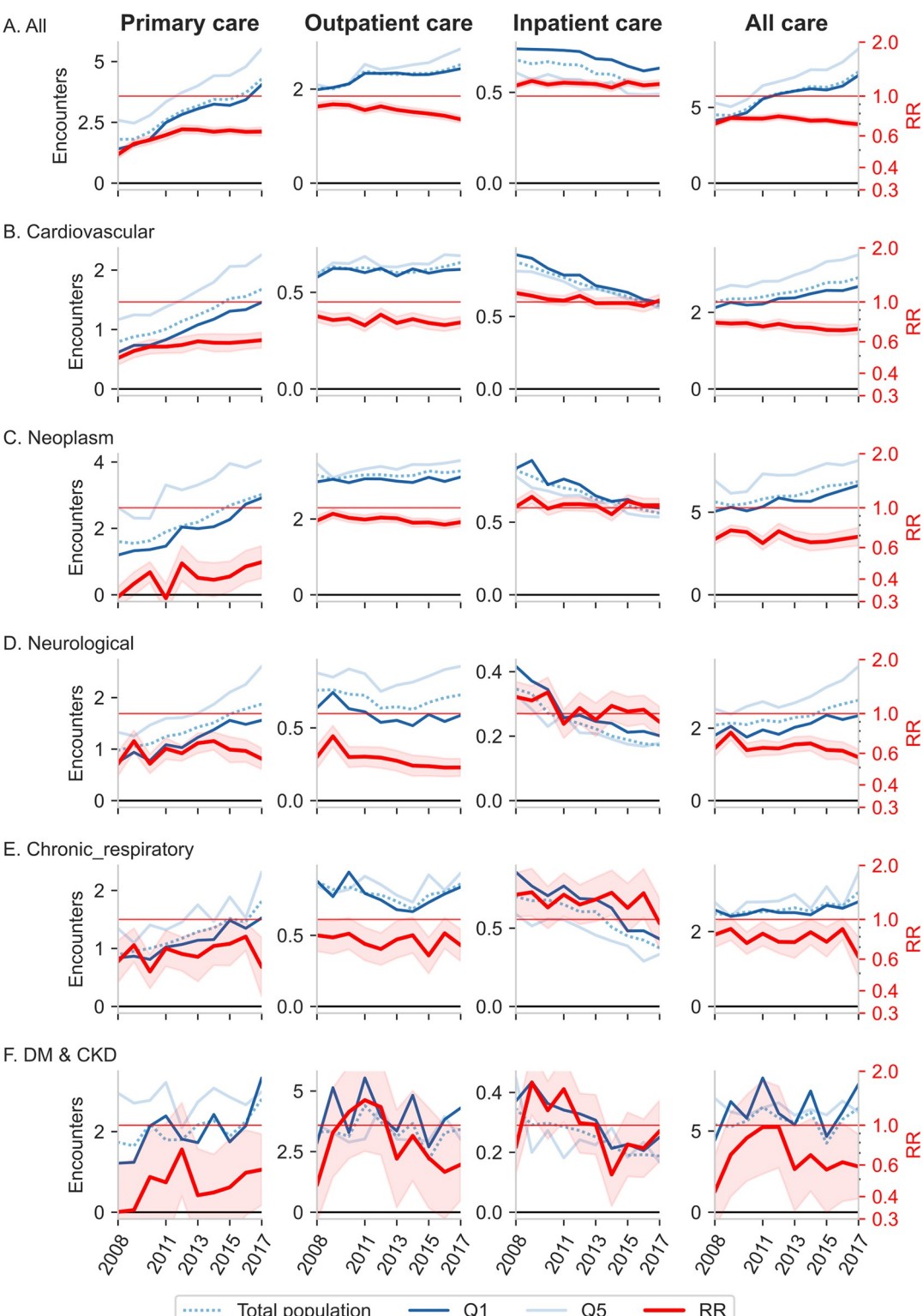

**Fig 4. Average number of healthcare encounters and adjusted rate ratios (RRs) ≤5 years prior to death.** The number of healthcare encounters by income group (Q1 and Q5) and for the total study population are displayed in blue. Adjusted RRs comparing Q1 with Q5 are in red, and shaded red areas denote 95% CI. RRs are adjusted for sex, age, age squared, country of birth, and civil status. Both adjusted and unadjusted results are available in S5 Table. Horizontal grid lines for RR = 1 (red) and encounters = 0 (black) are shown in the figures.

of primary and outpatient care as compared to mortality was observed for neoplasms. Overall, the large inequality in mortality rates were not reflected in correspondingly large group differences in HCU, and the discrepancy between large differences in mortality and lower in HCU was further widened each year.

A similar pattern was found when investigating the number of healthcare encounters ≤5 years prior to death: The largest group differences were observed in primary care, followed by outpatient care. For inpatient care, there were no significant differences. Again, group differences in primary care were largest for neoplasms, whereas group differences in utilization of outpatient care were largest for neurological disorders.

Although trend analyses of income-based HCU inequalities are sparse, our results are in line with previous cross-sectional survey studies. In Italy, Petrelli and colleagues found that high-income earners utilized both more primary and outpatient care, but less inpatient care, compared to low-income earners [16]. This pattern is well documented in a wide range of high-income countries [4,17,18], although there are a few older studies that only partly confirmed this [19,20].

Analogous to our findings, a longitudinal Norwegian survey study showed that socioeconomic differences in primary care utilization diminished over time between 1986 and 2006 [21]. Moreover, groups with high SES were more prone to utilize outpatient care, after adjustment of self-reported need. Finally, the income-based ratio of inpatient care was stable through the period, with considerably higher rates among low-income groups [21]. Similarly, in a Swedish sample from the years 1996/1997, Burström found an inverse income gradient related to having the need, but not seeking medical care [10].

This study adds to the previous literature by providing time trends in income-based differences in care utilization, stratified by disease groups. We observe a consistent and notable difference in rates of all-disease inpatient care and mortality between Q1 and the other income quintiles. Our results indicate that low-income earners tend to underutilize primary and outpatient care, especially for neoplasms and chronic respiratory diseases. Lower rates of healthcare for prevention and early treatments might contribute to higher rates of inpatient care. Meanwhile, low-income groups have higher HCU for neurological disorders that are less responsive to treatment. Further research exploring the benefits of allocating resources toward prevention and early treatments particularly for neoplasms and chronic respiratory diseases among low-income groups would be beneficial.

The importance of improving preventive and early care for low SES groups is also highlighted by a recent report on disparities in Swedish cancer care. The study revealed that individuals with lower education experienced delayed access to care, longer waiting times, and suboptimal treatments for various types of cancers, compared to individuals with higher education [22].

It is noteworthy that the association between SES and HCU in part depends on the operationalization of HCU. Lueckmann and colleagues [4] found that studies that defined HCU in terms of the probability of having visited a physician ("yes or no") reported larger inequalities in favor of high SES groups than studies reporting on frequencies (i.e., the number of healthcare encounters) or, more rarely, conditional frequencies (meaning the number of encounters given the individual had visited a physician at least once). Hence Lueckmann and colleagues suggested that HCU inequalities would best be tackled by removing barriers in initial access to specialized outpatient care for deprived groups [4]. In the current study, we found that low income was associated with both reduced probability and reduced frequency (in the last 5 years of life) of primary and outpatient care utilization.

The importance of adjusting for need when analyzing socioeconomic differences in HCU is widely acknowledged [23]. "Need" may be conceptualized as the capacity to benefit from care.

Need is commonly measured either as self-reported general health status or as self-reported presence of disease. Among the studies listed by Lueckmann, the majority employed the former. However, for population-wide register data, measures of subjective health are typically absent (as in [6]), which is also the case in our study. Instead, we compared income-related differences in HCU with group differences in mortality. Since we did not adjust income-related HCU differences for differences in need, they were likely due to a combination of differences in prevalence, access to care, and care-seeking behavior independent of prevalence. The relative importance of each of these factors is, however, unknown to us and likely play out differently in primary care, outpatient care, and inpatient care, respectively. Whereas income-related differences in utilization of primary and outpatient care likely are driven by differences in both prevalence and care-seeking behavior, income-related differences in inpatient care are probably driven almost exclusively by differences in prevalence of the conditions requiring hospital admission. This is further substantiated by the observed corresponding magnitudes of mortality differences. It is tempting to speculate that given perfect horizontal equity (i.e., equal treatment for equal need), any relative group differences in utilization rates of primary and outpatient care should be of similar magnitude as differences in inpatient use and mortality. However, the fact that we found substantially larger OR for mortality and inpatient care could also be due to reasons other than HCU inequalities. For instance, if low SES individuals would be more prone to more severe conditions (for instance, comorbidities) for which primary and outpatient care are not a primary point of entry, one would expect larger relative group differences in inpatient care and death. To better determine this, future studies should use appropriate control for group differences in medical need.

Despite not adjusting for HCU need, we can still draw important conclusions about the trends in HCU inequalities. If we assume a stable relationship between need and mortality throughout our study, the increasing discrepancy between income-related differences in HCU and mortality suggests an increase in HCU inequalities.

Several explanations for increasing HCU inequalities could be hypothesized. Although financial barriers to access to care have been largely removed in modern welfare states, the so-called *disproportionate care law* suggests that socially disadvantaged people receive more healthcare but of worse quality and insufficient quantity to meet their additional needs [2]. An increasing number of private health providers and the free right to establish care centers in any location has, in recent years, reduced the number of child and maternal care units in poorer areas where the need is greater [24]. There is also evidence of a preference for establishing healthcare facilities in high-income areas among private providers [25]. Also, the virtualization of primary care likely contribute to HCU inequalities, considering higher rates of digital HCU among individuals with higher education and income, even after adjusting for age and prevalence of chronic diseases [26]. In addition, availability is further skewed in favor of high SES HCU due to the increase in private insurances and out-of-pocket healthcare spending [27]. However, since our study is based on records of publicly financed healthcare usage, privately paid care is omitted from our analyses. If anything, including it would likely further accentuate the observed discrepancy between income-based differences in HCU and mortality.

Interestingly, in the sensitivity analyses in which we also adjusted for county effects, the ORs for primary care were higher, and income-related disparities in the number of primary healthcare encounters during the last 5 years of life were greatly diminished. The most significant reduction in group differences was observed for neoplasms. These findings align with a recent report that highlighted regional and socioeconomic disparities in Swedish cancer care, concluding that regional factors play a pivotal role in driving healthcare inequalities in cancer treatments [22].

To the best of our knowledge, this is the first study to present time series of income-related group differences in the utilization of primary, outpatient, and inpatient care alongside differences in mortality rates, covering the Swedish adult population for more than a decade. This was made possible through a unique, comprehensive database of linked microdata, which consisted of sociodemographic information, cause of death, and HCU both in specialized care and in primary care across most Swedish counties.

Most previous studies on SES differences in HCU rely on survey data. Although these provide valuable information on self-reported need of care, survey studies are typically based on just a small fraction of the general population, interviewed at only a few time points. National population registers, on the other hand, provide (near) full population coverage and enable unbiased definitions of income groups [13]. Importantly, the continuously recorded HCU register data enabled detection of time trends with high precision.

This study has several limitations. While it provides a comprehensive overview of time trends in income-related differences in HCU and mortality, it does not delve into a detailed analysis of specific patterns of group differences in HCU. A more nuanced understanding of these patterns would be valuable for identifying the mechanisms underlying avoidable health inequalities and to guide targeted interventions. For instance, pooling relatively heterogeneous diseases into broad disease categories masks the more extreme and possibly preventable income-related differences in HCU. Further, our metrics of HCU tell little about inequalities in the quality and content of the provided treatments.

Another limitation is the unknown and time-dependent coverage of primary care data. Coverage is likely higher in the more population-dense counties, which typically also coincide with higher income gradients and higher concentrations of high-income individuals. If true, this might partly explain the relatively high primary care utilization among high-income earners. However, similar patterns of high HCU among high-income earners are also observed in the outpatient care, for which the data coverage has been consistently high [28]. A national register of primary care data akin to what is available in, for example, Norway and Finland, would prove very valuable for future research.

Further studies should investigate socioeconomic differences in HCU patterns with higher resolution in terms of disease categories, treatments, and patient flows [29], for instance, by including income, HCU, and mortality in a common statistical model. Our study indicates that a promising target for reducing health (care) inequalities is to facilitate access to primary and outpatient care for low-income groups. Increased knowledge on how to promote care-seeking related to neoplasms, cardiovascular diseases, and chronic respiratory diseases in low SES groups would be particularly valuable, considering their high prevalence and the large discrepancies between income-related differences in HCU and income-related differences in mortality for these disease categories.

To conclude, in this study, we have identified major discrepancies between income-related differences in HCU and income-related differences in mortality, which widened with time. While high-income earners utilized an increasing share of primary and outpatient care, the inequality in mortality increased each year. This was especially noticeable for neoplasms. Facilitating motivated use of primary and outpatient care among low-income groups could help mitigate the growing health inequalities.

## Supporting information

**S1 Checklist. STROBE statement checklist.**
(DOCX)

**S1 Appendix. Supplemental methods.**
(DOCX)

**S1 Table. The year from which primary care data are available from each county, and contact to data sources.** Primary care data did not necessarily constitute the basis for disbursements, and the coverage rates could not be well verified and likely improved with time.
(DOCX)

**S2 Table. ICD definitions of disease groups.**
(DOCX)

**S3 Table. Annual rates (per 100,000) of diagnosed unique individuals and mortality, categorized by disease group, and data type, for the total population and for each income group (Q1 and Q5), alongside the mean age (± standard deviation).** (Since socioeconomic data were available only for fully survived calendar years, dates of death are projected to the previous year, hence rendering systematic underestimation of true age of death).
(XLSX)

**S4 Table. Adjusted and unadjusted ORs (Q1 vs. Q5) in 2017, 2004, and yearly change of log-transformed ORs.** [a]Model 1: Logistic regression testing for relative group differences (Q1 vs. Q5) controlling for age, age squared, sex, country of birth, and civil status. [b]Model 2: unadjusted. [c]Model 3: Same as Model 1 but also controlling for county. [d]P values associated with the OR indicate significant deviations from the null hypothesis of OR = 1.0, as calculated using Wald tests. [e]Beta is the estimated value of yearly change in log-transformed OR and is calculated by regressing log-transformed OR on time. The 95% confidence intervals (CIs) were obtained by bootstrapping method. CVD, cardiovascular diseases; Neuro., neurological disorders; Chro.Resp., chronic respiratory diseases; DM & CKD, diabetes and chronic kidney diseases; Inp., inpatient care; Outp., specialized outpatient care; Prim., primary care.
(XLSX)

**S5 Table. Income-based difference in number of disease-specific healthcare encounters the last 5 years of life. Adjusted and unadjusted RRs (Q1/Q5) in 2017, 2004, and yearly change of log-transformed RR.** [a]Model 1: negative binominal regression testing for relative group differences (Q1 vs. Q5) in number of healthcare encounters controlling for age, age squared, sex, country of birth, and civil status. [b]Model 2: unadjusted. [c]Model 3: same as Model 1 but also controlling for county. [d]P values associated with the OR indicate significant deviations from the null hypothesis of RR = 1.0, as calculated using Wald tests. [e]Beta is the estimated value of yearly change in log-transformed RR and is calculated by regressing log-transformed RR on time. The 95% confidence intervals (CIs) were obtained by bootstrapping. CVD, cardiovascular diseases; Neuro., neurological disorders; Chro.Resp., chronic respiratory diseases; DM & CKD, diabetes and chronic kidney diseases; Inp., inpatient care; Outp., specialized outpatient care; Prim., primary care.
(XLSX)

**S1 Fig. Rates (percents) of HCU and mortality by income group over time.** Q1, lowest income quantile; Q5, highest income quantile. All, all-disease; DM & CKD, diabetes and chronic kidney diseases.
(TIF)

**S2 Fig. Time courses (2004–2017) of rates (cases per 100) and adjusted ORs of HCU and mortality, using the same study population to analyze specialized care and mortality as was used for primary care.** This sensitivity analysis confirms the robustness of the results

shown in Fig 2 in relation to variations in the study sample. HCU (yearly rates of unique individuals) and mortality rates, by income group and for the total study population, are shown in blue colors. Rates are presented in percentages for display purposes. Adjusted ORs (95% CI) comparing Q1 with Q5 are in red, and shaded areas denote 95% CI. ORs are adjusted for sex, age, age squared, country of birth, and civil status. Red horizontal grid lines denote OR = 1, and the black grid lines denote the zero levels of rates. All, any ICD-10 code; CVD, cardiovascular diseases; Neuro, neurological disorders; Chro. Resp., chronic respiratory diseases; DM & CKD, diabetes and kidney diseases; HCU, healthcare utilization; OR, odds ratio.
(TIF)

## Acknowledgments

We are grateful to all providers of register data and, in particular, to the providers of regional primary care data. We also thank Hugo Sjöqvist, Lode Van Der Velde, and Dorien Beeres for valuable discussions on the data analyses, and Kasra Zarei for language proofreading.

## Author Contributions

**Conceptualization:** Pär Flodin, Peter Allebeck, Bo Burström, Emilie E. Agardh.

**Data curation:** Pär Flodin, Emilie E. Agardh.

**Formal analysis:** Pär Flodin.

**Funding acquisition:** Emilie E. Agardh.

**Investigation:** Pär Flodin.

**Methodology:** Pär Flodin.

**Resources:** Pär Flodin.

**Software:** Pär Flodin.

**Validation:** Pär Flodin.

**Visualization:** Pär Flodin.

**Writing – original draft:** Pär Flodin.

**Writing – review & editing:** Pär Flodin, Peter Allebeck, Ester Gubi, Bo Burström, Emilie E. Agardh.

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
