## [Editor Report · Decision Letter 0]

29 Mar 2023

Dear Dr Flodin, 

Thank you for submitting your manuscript entitled "Income-based differences in health care utilization in relation to mortality: Trends in the Swedish population between 2004-2017" for consideration by PLOS Medicine.

Your manuscript has now been evaluated by the PLOS Medicine editorial staff and I am writing to let you know that we would like to send your submission out for external peer review.

Please re-submit your manuscript within two working days, i.e. by Mar 31 2023 11:59PM.

Kind regards,

Philippa Dodd, MBBS MRCP PhD

PLOS Medicine

---

## [Decision Letter · Decision Letter 1]

25 May 2023

Dear Dr. Flodin,

Thank you very much for submitting your manuscript "Income-based differences in health care utilization in relation to mortality: Trends in the Swedish population between 2004-2017" (PMEDICINE-D-23-00836R1) for consideration at PLOS Medicine. 

[LINK]

In light of these reviews, I am afraid that we will not be able to accept the manuscript for publication in the journal in its current form, but we would like to consider a revised version that addresses the reviewers' and editors' comments. Obviously we cannot make any decision about publication until we have seen the revised manuscript and your response, and we plan to seek re-review by one or more of the reviewers. 

We expect to receive your revised manuscript by Jun 15 2023 11:59PM. Please email us (plosmedicine@plos.org) if you have any questions or concerns.

We look forward to receiving your revised manuscript. 

Sincerely,

Philippa Dodd, MBBS MRCP PhD

PLOS Medicine

plosmedicine.org

GENERAL

Please respond to all editor and reviewer comments detailed below in full.

Please ensure that the study is reported according to the STROBE guideline, and include the completed STROBE checklist as Supporting Information. Please add the following statement, or similar, to the Methods: "This study is reported as per the Strengthening the Reporting of Observational Studies in Epidemiology (STROBE) guideline (S1 Checklist)."

When completing the checklist, please use section and paragraph numbers, rather than page (and/or line) numbers as these often change in the event of publication.

DATA AVAILABILITY STATEMENT

Please include an appropriate email or web address for the Swedish National Board of Health and Wellfare and Statistics Sweden.

COMMENTS FROM THE ACADEMIC EDITOR

This is an interesting paper and I agree with a major revision. I agree that adjusting for baseline health status is important. Reviewer 1 in particular has very helpful input. One of my concerns is the lack of a conceptual theory or model here. What are we primarily interested in--utilization or outcomes? What is the main relationship they are hypothesizing for these two--that the poor should have more utilization? Of what kinds of services? They are missing a key reference which might help them frame this better: Cookson R, Doran T, Asaria M, Gupta I, Mujica FP. The inverse care law re-examined: a global perspective. Lancet. 2021 Feb 27;397(10276):828-838. doi: 10.1016/S0140-6736(21)00243-9. PMID: 33640069.

ABSTRACT

Abstract Background: 

Please expand details to provide wider context to the reader.

Line 15 -you state the study population included >15 year olds but in the main methods (line 81) you state 16 years – please clarify/revise for consistency.

Please move details of your study population to the methods and findings section of your abstract.

The final sentence should clearly state the study question.

Abstract methods and findings:

Line 19 – please include some of the sociodemographic variables to which you refer. 

Please include any important variable factors that are adjusted for in your analyses.

Please clearly define the setting (national data?), number of participants whose data you include, and the main outcome measures.

Please quantify the main results with 95% CIs and p values. When reporting p values please report as p<0.001 and where higher as p=0.002, for example. Suggest reporting statistical information as follows ‘(OR 1.78, 95% CI [1.74,1.82[; p</=)’.

AUTHOR SUMMARY

At this stage, we ask that you include a short, non-technical Author Summary of your research to make findings accessible to a wide audience that includes both scientists and non-scientists. The authors summary should consist of 2-3 succinct bullet points under each of the following headings:

• Why Was This Study Done? Authors should reflect on what was known about the topic before the research was published and why the research was needed.

• What Did the Researchers Do and Find? Authors should briefly describe the study design that was used and the study’s major findings. Do include the headline numbers from the study, such as the sample size and key findings. 

• What Do These Findings Mean? Authors should reflect on the new knowledge generated by the research and the implications for practice, research, policy, or public health. Authors should also consider how the interpretation of the study’s findings may be affected by the study limitations. In the final bullet point of ‘What Do These Findings Mean?’, please describe the main limitations of the study in non-technical language.

The Author Summary should immediately follow the Abstract in your revised manuscript. This text is subject to editorial change and should be distinct from the scientific abstract. Please see our author guidelines for more information: https://journals.plos.org/plosmedicine/s/revising-your-manuscript#loc-author-summary

METHODS and RESULTS

Did your study have a prospective protocol or analysis plan? Please state this (either way) early in the Methods section.

For all observational studies, we ask the authors to indicate the following in the manuscript text: 

(1) the specific hypotheses you intended to test, 

(2) the analytical methods by which you planned to test them, 

(3) the analyses you actually performed, and (4) when reported analyses differ from those that were planned, transparent explanations for differences that affect the reliability of the study's results. If a reported analysis was performed based on an interesting but unanticipated pattern in the data, please be clear that the analysis was data-driven.

As above (see under ABSTRACT), please quantify main results with 95% CIs and p values. When a p value is given, please specify the statistical test used to determine it.

When reporting p values please report as p<0.001 and where higher as p=0.002, for example. Suggest reporting statistical information as follows ‘(OR 1.78, 95% CI [1.74,1.82[; p</=)’

Line 93 – please change ‘parenthesis’ to ‘parentheses’

Line 94 - Why are diabetes and CKD grouped together? Do these data refer to those with diabetic kidney disease or those with diabetes and separately kidney disease of any other cause? Or does it represent 2 groups of individuals some with diabetes and some with CKD? Please clarify/justify for the reader as it is currently unclear.

Line 152 – ‘(OR=1.04, 95% CI:1.04-1.05)’ please use commas to separate upper and lower CI bounds as hyphens can be confused with reporting of negative values (and to ensure consistency of statistical reporting throughout your manuscript).

TABLES

Table 1 – presents very little data regarding the study population only total numbers and age, please include a comprehensive table detailing wider demographic details of the full cohort.

FIGURES

Please ensure that all figures are affiliated to an appropriate caption which clearly describes the figure content without the need to refer to the text.

Please consider avoiding the use of red and/or green to improve accessibility of your figures to those with colour blindness.

As above, when reporting 95% CIs please also report p values. Please report as p<0.001 or as p=0.002, for example. 

Figure 1 – please indicate in the figure caption which factors are adjusted for. Where adjusted analyses are presented, please also include unadjusted analyses to help facilitate transparent data reporting. Please indicate the meaning of the black lines.

Could the chart be slightly more clearly/accurately labelled – DM & CKD, for example

Figure 2 – what is the meaning of the shaded areas behind the red lines? What is the meaning of the dashed lines? What is the meaning of the horizontal thin red line? Please define for the reader. Please ensure consistent use of number formatting (for example, use of decimal places 5.0 Vs 5).

Figure 3 – why does it differ in colour coding when compared to figure 1 despite detailing the same categories? Please revise for consistency (ideally avoiding the use of red and/or green).

Figure 4 – In the figure title please revise to read ‘Healthcare utilisation (HCU)…’. What do the different (and sometimes overlapping) shaded areas represent? Please clearly define for the reader.

DISCUSSION

Please remove all sub-headings from the discussion such that it reads as a single piece of continuous prose. Please ensure that it is structured as follows: a short, clear summary of the article's findings; what the study adds to existing research and where and why the results may differ from previous research; strengths and limitations of the study; implications and next steps for research, clinical practice, and/or public policy; one-paragraph conclusion. Again, please avoid the use of sub-headings.

Line 213 – how is ‘severe’ defined and by whom?

SUPPORTING INFORMATION

TABLES

As above, please ensure that all tables (and figures) are affiliated to an appropriate caption that clearly describes its content without the need to refer to the text.

Please include p values where 95% CIs are reported (if not please clearly state the reasons why not).

Please indicate if your analyses are adjusted or unadjusted and where you present adjusted analyses, please clearly state the factors adjusted for. Please also include unadjusted analyses for comparison.

S3 Table – please see reviewer #1 comments which we agree with. Please revise accordingly to improve clarity.

S5 Table – in the column headers please define the numerical within and outside of parentheses.

REFERENCES

For in-text reference callouts, please place citations in square parentheses. 

Please see our website for other reference guidelines https://journals.plos.org/plosmedicine/s/submission-guidelines#loc-references

Comments from the reviewers:

Reviewer #1: General comment 

The paper focuses on income-based differences in healthcare utilization (HCU). Using a repeated cross-sectional register-based study design, authors analyze the relationship between income - comparing first and fifth quintile - and four dimensions (primary care, outpatient, inpatient, death) of healthcare use, controlling for main (five) diseases groups. 

The topic of the paper and methodological approach is interesting; however, it presents a number of shortcomings in many ways.

The most serious drawbacks are the following. 

First, authors include in their analysis both HCU and mortality, and this generate confusion about what is the topic that they are interesting to study. Analyzing the relationship between income and HCU or mortality need different approaches, theories, and scientific literature references. Furthermore, from reading the initial part of the paper it would seem the authors intend to study the relationship between HCU and mortality in a single model, on contrary they are treated as distinct outcomes. Looking at HCU in the five years prior to death (question 3: How do the low- and high-income groups differ in numbers of health care encounters ≤5 years prior to death?) does not justify the inclusion of mortality as a topic of the paper (questions 1 and 2: 1. How do low- and high-income groups differ with regard to HCU and mortality in 2017 and over time? 2. To what extent do income-related differences in HCU and mortality vary across disease groups?). At page 15, rows 282-283, authors' statement "…explaining the origin of the observed increase in mortality inequalities is challenging, and differences in HCU likely only play a partial role" it is probably true, but authors did not test this hypothesis in the paper. Moreover, pag. 13, rows 217: "…the large inequality in mortality rates were not reflected in correspondingly large group differences in HCU". To do it, they should have adopted a different analytical strategy: evaluate the impact of SES and HCU on mortality in a single model (for example using a SEM or GSEM models) to estimate direct and indirect effects.

Second, as the authors themselves acknowledge, it is important to adjust for need when analyzing socioeconomic differences in HCU. The lack of this control variable in the models makes the estimates (differences between income quintiles) predictably biased. Treating the differences in mortality as a proxy for disparities in needs can not be considered an acceptable analytical strategy, since it uses an outcome, i.e. mortality, which is also the result of different needs related to socioeconomic inequalities to control for differences in health needs. This approach is tautological. 

Third, authors produce separate estimates for the main disease groups that account for the largest number of deaths in Sweden in 2017. This is certainly interesting, but they did not explain the differences observed in the results between the diseases groups, reporting only the estimated values. So, what is the utility of doing these analyses?

Fourth, the Discussion paragraph is very fragmented, making it difficult to understand if and to what extent the hypotheses are corroborated and how to interpret the results in the light of the aims of the paper. For example, if the theoretical background of the paper is the fundamental cause theory, it should be introduced in the first part of the paper and the results should be discussed in the light of this theory. 

Other shortcomings

Pag. 5. The study population comprised all individuals 16 years and older, but, as we know, most young people enjoy good health, as we can also see from the average age in the use of different types of health services (table S3), therefore the analyzes could be conducted on a population with a different age range.

Pag. 6. Authors should discuss the reliability of the income data used dataset, as we know this kind of data is difficult to collect with accuracy.

Pag.13, rows 212-213. Authors state: "For all diseases combined, the 213 more "severe" the outcome, the larger the inequalities". How is the degree of severity of the health service used established?

Pagg. 16-17. Authors did not clearly explain because in a universal healthcare they observe differences in HCU on the basis of income levels. For example, they wrote: "Although financial barriers to access to care have been largely removed in modern welfare states, the so-called 300 inverse care law suggests that "availability of good medical care tends to vary inversely with the need of the population served" ([1] p 459, Hart 1971). An increasing number of private health providers and the free right to establish care centres in any location has in recent years reduced the number of child and maternal care units in poorer areas where the need is greater (24). In addition, availability is further skewed in favour of high SES HCU due to the increase in private insurances and out of pocket health care spending". First, why should people take out private insurance in a universal healthcare system like the Swedish one? This point should be further explored. Second, authors did not control for these possible confounding variables in their analysis. Third, examples about the inverse care low refer to healthcare services which are not related to groups of diseases considered in the paper. 

Pag. 22. S3 Table caption is partial, and content is not clear. For example, what does Full population, N in the fourth column mean? The content is not an absolute value but a rate, therefore N is wrong here.

Reviewer #2: The manuscript describes income differences in health care utilisation (HCU) and mortality in Sweden between 2004-2017, and the development in these over time. It finds small differences by income in primary and outpatient care, but large and rising inequalities in inpatient care and mortality. They also display interesting differences in the social gradient in HCU and mortality by disease group for five major groups. All in all, I found the manuscript an interesting read which should be relevant for a broad audience. Here are a few comments.

1. Minor points on the study population: The population specification in line 81 should be residents in 2004-2017, a typo. Table 1 should have a title that describes its content, i.e. size and age of study population.

2. They utilise primary care data that they have collected themselves from counties, as there is no central register. This is a major endeavour and contributes to the novelty of the study, which could perhaps be highlighted in the introduction; i.e., due to unique data, this is the first study to consider income differences in primary care use in Sweden and compare it to other HCU. Given this novelty, the highly interested reader would also want to know more about this dataset, perhaps in the supplementary material, including whether you believe it is reliable/complete and whether it is the basis for disbursements. Especially, I am curious which counties contributed how many years, and which did not. The authors have anonymised this information, which is a shame for both replication purposes and interpretation, because Swedish counties are very different wrt. degree of centrality, immigration, health care availability and inequality. If anonymisation is a requirement, that should be stated. They should include some summary statistics showing whether the included county-years differ from the total population (on these dimensions, and also in levels and differences in specialist HCU and mortality). They should display the main results (e.g., Fig. 4) with the same county-years as their primary care sample for the other outcomes, possibly as supplementary material. Furthermore, they should return to this in the discussion. A strength is that they look beyond primary care. A weakness is that primary care data coverage is incomplete, they should speculate on how that may have affected the results, and optionally call for a national register which could improve coverage and data quality for future research. Swedish policy-makers need to hear this from the researchers, it is one of the few aspects where Swedish data is not to the standard of other Nordic countries.

3. Relatedly, in their data statement they state that «Data are available from the Swedish National Board of Health and Welfare and Statistics Sweden». Does this also cover the collected primary care data? What are the prospects of also making that data available for other researchers? Could you give a reference number for the data orders to the National Board and Statistics Sweden, so that other researchers with the necessary permits could order the same?

4. I find the methods section under-explained and at times confusing, and suggest a larger methods section, perhaps in the supplementary materials. Ideally, the methods should be well enough described for other researchers to be able to replicate the study and assess the comparability with other studies. I encountered an empty repository when following the provided link. Seeing the code could have helped, but ideally the manuscript should be self-explanatory without needing to read code. They have used what they refer to as family income. It is not clear what constitutes a family and why household income was not used, which limits comparability to e.g. Chetty et al. (2016) and Kinge et al. (2019). The equivalence scale used should be specified. Equations used for the regression analyses should be stated. In the logistic regression, the method for computing standard errors is not specified, clustering on individuals would be a reasonable choice since the same person is observed multiple years. It is unclear how they regressed log transformed ORs on time, whether they used individual year dummies or parametric time trends.

5. I would read this paper with an interest in income differences in the entire Swedish population. However, the paper quickly develops into a comparison of two outlying groups, Q1 vs. Q5. Instead of Fig 1, I would want to see the ORs for all groups Q1-Q4 compared to Q5. In Fig 2, I would want to see the development in all five quintiles. It is a very different inequality if all groups Q2-Q5 have similar levels and Q1 is an outlier than if it is Q5 that differs from the rest. It is also important for our understanding how each group develops across time.

6. I would encourage the authors to rely on their graphic depictions of each quintile's development across time, rather than running the log-transformed time series analysis. The additional analysis seems unnecessarily complicated, and again narrows the focus to Q1 vs Q5 developments. I also struggle to trust and interpret the estimates. Take their estimate for all diseases, they find a yearly change in log ORs of -0.006 for outpatient care, and 0.012 for inpatient care (see table S4). This is a complicated size, they do not provide any direct interpretation or unit of measurement. It could seem like the negative change in outpatient care (i.e. decreasing inequality) is half the positive change in inpatient care (i.e. increasing inequality). However, when eyeballing Fig. 2, I see a clear increase in inequality in inpatient care and a bumpy but flat development in inequalities in outpatient care. In fact, in table S4 we can also read that the Q1/Q5 OR slightly increased from 1.038 to 1.042 in 2004-2017 in outpatient care, although the time trend estimate is a precise and sizeable negative. 

7. Figure 4 is difficult to appreciate. It is not clear why the authors are now also interested in Q3, I would again like to see all quintiles. They have changed the reporting to percentage change, however I suspect the estimates are still based on ORs. They lack a justification for why Fig 4 is displayed differently than Fig 2, having the same presentation on this outcome would make the reading easier. This should be the absolute development of all quintiles, and if they wish, add the Q5 vs Q1 OR comparison. Having all the scales going in +/- 50% makes the figures less readable. In general, their conclusion is that they don't find significant income differences in primary or inpatient care in any disease group 5 years before death, but sizeable under-utilisation of outpatient care for most disease groups. Also displaying overall HCU 5 years prior would make this point more clearly, and improve power (which seems to be a problem in this analysis). This finding should be picked up more in the discussion. What characterises outpatient care in Sweden, why is it so under-used by low-income groups with short remaining life expectancies, and could improving outpatient HCU reduce mortality for low-income groups?

8. In the discussion, the causes and consequences of differences in utilisation by disease group seems underdeveloped. To me, this is one of the most striking findings in the paper. Low-income groups seem to under-utilise primary and outpatient care for neoplasms and chronic respiratory diseases, but use them frequently (and increasingly so) for neurological diseases and diabetes. They see the doctor, but perhaps not for the right reason. A pattern emerges where these groups use health care too little for prevention, resulting in an over-usage for treatment. Could GPs use the consultations with these groups that are requested for diabetes and neurological disorders to prevent and detect cancer and COPD risks, and through that possibly affect later need for inpatient care and mortality risk? This could be one point for future studies.

9. In the discussion on causes of increasing inequality in mortality, the authors write that inequalities tend to increase with declining overall mortality levels (lines 272-273). The authors should add an explanation of why it is the case that inequality increases as mortality declines, and perhaps discuss whether this holds in all stages of the epidemiologic transition, including the fourth stage of decline in degenerative diseases (Ohlsansky & Ault, 1986). The main point in their reference (Mackenbach, 2019) is that SES differences tend to remain (not increase) even as overall mortality declines, but also that there are large variations in this relationship between countries and across time periods. Increased socio-economic selection is one factor he mentions that may cause this, as social mobility and increasing returns to education makes income more predictive of general skills and abilities. Another is that taking advantage of improvements in public health requires ability for behavioural adaptation and health literacy. Some explanation/moderation is needed so that the co-development does not appear to be deterministic.

10. The title of Table S3 suggests that it presents rates, but it also presents average ages. The ages can be dropped or need to be stated in the title. In the column headings, they use N for rates which is unfortunate; write out rate instead. N can easily be confused with number of observations, and they use N for this in Table 1. 

11. I would be interested in all ORs Q1-Q4 in Table S4 and S5, not only Q1 vs. Q5 and Q1 and Q3 vs. Q5 respectively.

References:

Chetty, R., Stepner, M., Abraham, S., Lin, S., Scuderi, B., Turner, N., ... & Cutler, D. (2016). The association between income and life expectancy in the United States, 2001-2014. Jama, 315(16), 1750-1766.

Kinge, J. M., Modalsli, J. H., Øverland, S., Gjessing, H. K., Tollånes, M. C., Knudsen, A. K., ... & Vollset, S. E. (2019). Association of household income with life expectancy and cause-specific mortality in Norway, 2005-2015. Jama, 321(19), 1916-1925.

Mackenbach, J. P. (2019). Health inequalities: Persistence and change in modern welfare states. Oxford University Press, USA.

Olshansky, S. J., & Ault, A. B. (1986). The fourth stage of the epidemiologic transition: the age of delayed degenerative diseases. The Milbank Quarterly, 355-391.

Reviewer #3: By linking multiple datasets (Longitudinal Integrated Database for Health Insurance and Labour market Studies, National Patient Register, Cause of Death Register), the authors tried to assess time trends of income-related differences in use of different levels of health care in relation to mortality between the years 2004 and 2017. The three research questions of the study are 1)

How do low- and high-income groups differ with regard to HCU and mortality in 2017 and over time? 2) To what extent do income-related differences in HCU and mortality vary across disease groups? 3) How do the low- and high-income groups differ in numbers of health care encounters ≤5 years prior to death? Overall, I think the study topic is of importance and the results are relevant for policy making. However, I think the study methods would require some improvements. Below are my specific comments. 

1. The data are not fully publicly available, which may not comply with PLoSMed's policy. 

2. Line 47, and 53-55, instead of quoting the original sentences, I think it would be better to paraphrase the original sentences or summarize the main idea of the original sentences, especially when the quoted sentences are very long. 

3. Line 43-78: The introduction could benefit from providing more context on the global issue of SES disparities and their effects on health care access and outcomes. This would allow readers to better understand the broader implications of the study.

4. Line 81, between 2004 and 2017?

5. Line 108-110, are the income metrics listed here comprehensive and representative of people's wealth? I am thinking real estate properties people have or other capital assets and may not yield capital return. 

6. Line 113. Also, I think income level is only one of the many contributing factors to SES level. Income level cannot even represent people's wealth accurately. Social status (such as job type) and education level are also important contributing factors to SES level. I think it's a bit inaccurate to use income alone to define SES. I would suggest the authors just use income level instead of SES level. 

7. Line 126-127, I think the model also needs to adjust for people's health status or perceived health status because that can be an important bias covariate of the relationship between income level and HCU or death. 

8. Figure 2: The y-axis on the left should be percent or rate? Percent and rate are quite different concepts.

[LINK]

---

## [Decision Letter · Decision Letter 2]

29 Aug 2023

Dear Dr. Flodin,

Thank you very much for submitting your manuscript "Income-based differences in health care utilization in relation to mortality: Trends in the Swedish population between 2004-2017" (PMEDICINE-D-23-00836R2) for consideration at PLOS Medicine. 

[LINK]

In light of these reviews, I am afraid that we will not be able to accept the manuscript for publication in the journal in its current form, but we would like to consider a revised version that addresses the reviewers' and editors' comments. Obviously we cannot make any decision about publication until we have seen the revised manuscript and your response, and we plan to seek re-review by one or more of the reviewers. 

We expect to receive your revised manuscript by Sep 19 2023 11:59PM. Please email us (plosmedicine@plos.org) if you have any questions or concerns.

We look forward to receiving your revised manuscript. 

Sincerely,

Philippa Dodd, MBBS MRCP PhD

PLOS Medicine

plosmedicine.org

GENERAL

Thank you for your detailed responses to previous editor and reviewer comments. Please see below for further comments which we require you address in full.

Throughout, where possible including in the figure and table captions please refrain form using the word disease (exception would be the ICD-10 coding) and replace with an alternative, perhaps co-morbidity or similar.

*** Reviewer #2 has asked for the inclusion of additional analyses (please see below) which have the potential to alter the interpretation of your data and for that reason we have requested a further major revision ***

DATA AVAILABILITY STATEMENT

Thank you for including contact details, as per my previous request, in the main manuscript for Swedish National Board of Health and Welfare and Statistics Sweden. Please only include these in the manuscript submission form and remove from the manuscript itself. 

Please also place the details you provide in table S1 in the relevant part of the manuscript submission form. Please indicate that these are for access to primary care data, as we understand things.

In the event of publication these details will be compiled as metadata.

In the manuscript submission form, please replace ‘Wellfare’ with ‘Welfare’.

COMPETING INTERESTS

All authors must declare their relevant competing interests per the PLOS policy, which can be seen here:

https://journals.plos.org/plosmedicine/s/competing-interests

For authors with ties to industry, please indicate whether any of the interests has a financial stake in the results of the current study.

ABSTRACT

In the last sentence of the Abstract Methods and Findings section, please describe the main limitation(s) of the study's methodology.

AUTHOR SUMMARY

Thank you for including an author which reads very nicely.

Line 56 – suggest ‘data’ instead of ‘microdata’

Line 59 – not sure ‘diminishing’ is the right word here, it makes the sentence a little unclear. Do you mean that despite having the highest mortality rates, those in the lowest income quintile access healthcare the least? Please revise for clarity.

Line 63 – what do these findings mean? - please describe the main limitations of the study in non-technical language. ‘lack of adjustment for individual healthcare need’ at line 64 could form part of this point.

METHODS

Line 189 – please remove the funding statement and include only in the manuscript form when you resubmit the manuscript. In the event of publication it will be compiled as metadata.

Lines 191 -193 (STROBE statement) please move to the beginning of the methods section.

TABLES

Table 1 – characteristics of the study population still lacks adequate sociodemographic details which should be available when considering the data bases used. It is also not very accessible and presents selected data and not complete data. Please revise. 

Please review our currently published articles on our website https://journals.plos.org/plosmedicine/ for suitable examples. 

FIGURES

Please avoid the use of the color red and/or green to make your figures accessible to those with color blindness.

Figure 1 – please define OR in the caption ‘Adjusted odds ratio (OR)…’

Figure 2 - please define OR in the caption ‘Adjusted odds ratio (OR)…’

Figure 3 - please define OR in the caption ‘Adjusted odds ratio (OR)…’ and as per my previous request please define all abbreviations used to categorise co-morbidities. 

DISCUSSION

*** The language and grammar of the discussion needs careful attention as many sentences are unclear and appear partially written. Please check carefully throughout and amend as necessary. Specific points are detailed below but the list is no exhaustive ***

Line 312 – please remove ‘treatment in’

Line 313 – suggest ‘Further research exploring the benefit of allocating resources towards prevention and early treatment…would be beneficial.’ Or similar.

Line 315 – ‘rather than on less treatable neurological disorders.’ This could be misinterpreted and cause offense as it suggests neurological disorders aren’t worth investment. Please revise/remove.

Line 355 – ‘Despite we did not adjust HCU for need…’ please revise as this is grammatically incorrect, perhaps instead, ‘Despite not adjusting for HCU need…’ as we understand things.

Line 356 – ‘Assumed the existence of any…’ it is unclear what you are trying to say here, please revise for clarity 

Line 353 – ‘To better determine this, future studies should use proper control for group differences in medical need.’ Please use an alternative word to ‘proper’ which is unscientific and vague.

Line 426 – as above, please remove the data availability statement from the main manuscript and include only in the manuscript submission form. In the event of publication it will be compiled as metadata.

REFERENCES

For web references please replace the term ‘cited’ with ‘Accessed’

SUPPORTING INFORMATION

S4 and S4 Tables – thank you for including these tables which are very clear. Please also define ‘confidence interval (CI)’ in the footnotes of both.

S1 Fig – as above please refrain form using red and/or green to improve accessibility of your figures to those with color blindness.

Comments from the reviewers:

Reviewer #2: Major comment: 

The authors have responded adequately to my comment 1 and to some of the issues raised in my comment 2. However, I wanted to see the results on levels and trends in HCU and mortality for the same sample of counties where they have primary care data throughout the period. Simply put, the trends you see in primary care usage may be due to sample changes rather than a development in health care usage, unlike the specialist care case where statistics from the whole country are available throughout. The difference in levels between primary and specialist care seem to be partially due to the county composition, as controlling for county has an impact. They commented on that this matters when considering encounters during the last 5 years prior to death, but also in the main analysis, the 2004 estimate for primary care changes from 1.11 when not adjusting for county to 1.30 when adjusting, so that the Q1/Q5 ratio is actually higher than for inpatient care that year (where it is 1.22). These results from model 3 in S4 need to be discussed. It is of primary interest for the research question whether the difference they see between the income gradients in primary care and inpatient care use is in fact an artifact of sampling differences.

Minor comments: 

I commend the authors for thorough responses to my comment 3 and 4, and especially the updated method section and supplement S1 gave a much clearer picture of what they did. Now that I understand that they have repeatedly simulated linear trends, a question that I wish they could address in the paper is whether the developments are well proxied by linear trends. Stating that they have looked at the plots year-by-year and visually inspected that it looks linear could suffice.

In their response to comment 5, I find the figure for HCU and mortality rates for each income quintile over time to be very interesting, referred to as Figure S2 by the authors but appearing as S1 in the document. For inpatient care and deaths, there seems to be an increasing gap between Q1 and Q2-Q5 rather than a broader increase in income-related inequality. I believe it is worth reflecting on this in the discussion. Pointing out that results are driven by the poorest group falling behind would help the reader correctly interpret the Q1/Q5 results. I recognise that the sole focus on lowest and highest group is stated in the abstract and introduction, however the title promises income-based differences in the Swedish population, the majority of which are in Q2-Q4.

I would like to thank the authors for clarifying the estimates in response to comment 6, I now understand that the trend is a coefficient and thus on a different scale than the ORs.

Responding to comment 7, I believe FIgure 4 is now clearer and more consistent with the rest of the paper.

The comments 8-11 are adequately responded to.

Reviewer #3: I think the authors have adequately addressed the questions/concerns raised by me and by the other reviewers. I think the paper is acceptable in its current form. Congratulations to the authors.

[LINK]

---

## [Decision Letter · Decision Letter 3]

9 Oct 2023

Dear Dr. Flodin,

Thank you very much for re-submitting your manuscript "Income-based differences in health care utilization in relation to mortality: Trends in the Swedish population between 2004-2017" (PMEDICINE-D-23-00836R3) for review by PLOS Medicine.

I have discussed the paper with my colleagues and it was also seen again by 2 reviewers. I am pleased to say that provided the remaining editorial and production issues are dealt with we are planning to accept the paper for publication in the journal.

[LINK]

We look forward to receiving the revised manuscript by Oct 16 2023 11:59PM.   

Sincerely,

Philippa Dodd, MBBS MRCP PhD

PLOS Medicine

plosmedicine.org

Requests from Editors:

GENERAL

Thank you for your detailed responses to previous editor and reviewer comments. Please see below for further comments which we require you address, in full, prior to publication.

TITLE

Please revise the title to read, 'Income-based differences in health care utilization in relation to mortality in the Swedish population between 2004-2017: An observational cohort study.'

TABLES

Table 1 – we asked previously for inclusion of a table with more detailed demographic data. Thank you for your response. Apologies for the lack of clarity. The editorial team have discussed this point and agree that it is important for the reader to appreciate how the population studied is representative of the nation and how that may have changed over time. Considering that a lower proportion of people in Q1 were born in Sweden, we think this is important detail.

Please include an updated table including baseline demographic details – age (including range), ethnicity, income group, marital status, sex etc.

Please ensure that percentages are quantified with numerators and denominators.

This is a prerequisite to publication. We encourage you to refer to currently published articles on our website https://journals.plos.org/plosmedicine/ for suitable examples.

Comments from Reviewers:

Reviewer #2: All points raised have been responded to and I have no further comments.

Reviewer #3: I think the authors have adequately addressed the questions/concerns raised by the reviewers. I think the paper is acceptable in its current form.

[LINK]

---

## [Editor Report · Decision Letter 4]

12 Oct 2023

Dear Dr Flodin, 

On behalf of my colleagues and the Academic Editor, Professor Margaret Kruk, I am pleased to inform you that we have agreed to publish your manuscript "Income-based differences in health care utilization in relation to mortality in the Swedish population between 2004-2017: A nationwide register-based study" (PMEDICINE-D-23-00836R4) in PLOS Medicine.

Thank you for your detailed clarifications to our previous requests which we accept. The modified title is perfectly suitable. 

PRESS

Thank you again for submitting to PLOS Medicine. IT has been a pleasure handling your manuscript. We look forward to publishing your paper. 

Kind regards,

Pippa 

Philippa Dodd, MBBS MRCP PhD 

PLOS Medicine

pdodd@plos.org

plosmedicine@plos.org